# Metabolite Changes in Maternal and Fetal Plasma Following Spontaneous Labour at Term in Humans Using Untargeted Metabolomics Analysis: A Pilot Study

**DOI:** 10.3390/ijerph16091527

**Published:** 2019-04-30

**Authors:** Katherine A. Birchenall, Gavin I. Welsh, Andrés López Bernal

**Affiliations:** 1Department of Obstetrics and Gynaecology, St Michael’s Hospital, Bristol BS2 8EG, UK; ALopezBernal@bristol.ac.uk; 2Translational Health Sciences, University of Bristol, Bristol BS1 3NY, UK; G.I.Welsh@bristol.ac.uk

**Keywords:** human parturition, endocannabinoid, steroids, energy metabolism

## Abstract

The mechanism of human labour remains poorly understood, limiting our ability to manage complications of parturition such as preterm labour and induction of labour. In this study we have investigated the effect of labour on plasma metabolites immediately following delivery, comparing cord and maternal plasma taken from women who laboured spontaneously and delivered vaginally with women who were delivered via elective caesarean section and did not labour. Samples were analysed using ultra high-performance liquid chromatography-tandem mass spectrometry. Welch’s two-sample t-test was used to identify any significant differences. Of 826 metabolites measured, 26.9% (222/826) were significantly altered in maternal plasma and 21.1% (174/826) in cord plasma. Labour involves changes in many maternal organs and poses acute metabolic demands in the uterus and in the fetus and these are reflected in our results. While a proportion of these differences are likely to be secondary to the physiological demands of labour itself, these results present a comprehensive picture of the metabolome in the maternal and fetal circulations at the time of delivery and can be used to guide future studies. We discuss potential causal pathways for labour including endocannabinoids, ceramides, sphingolipids and steroids. Further work is necessary to confirm the specific pathways involved in the spontaneous onset of labour.

## 1. Introduction

Our understanding of the mechanisms of human parturition remains incomplete and consequently management options for complications of pregnancy such as preterm labour (PTL), and common interventions such as induction of labour (IOL) [1,2], remain clinically unsatisfactory [3].

A better understanding of human labour at term (defined as 37–42 weeks’ gestation) would allow development of methods for improved prediction and prevention of PTL, as well as more effective strategies for IOL [4,5,6]. The aetiology of PTL is heterogeneous and frequently pathological, however it may result in stimulation of the same triggers for parturition as spontaneous labour at term [1,7]. The proportion of pregnancies with IOL in England has increased from 20.3% in 2006–2017 to 29.4% in 2016–2017 [4,6,8,9,10]. IOL can be inefficient, with potentially severe complications for the mother and neonate, including lengthy hospital stays at high cost to the National Health Service and a negative impact on the birth experience [3,11,12]. The use of clinical and molecular markers to identify women at risk of complications of pregnancy remains challenging due to their poor predictive value [13,14]. There is a need to identify the metabolic changes underlying the physiological onset of spontaneous human labour and to develop new clinically useful predictive markers and better drug targets.

Metabolomics allows the investigation of a wide range of metabolites across the main biochemical pathways [15]. The advantage of studying the metabolome or metabolic profile of a system is that it reveals the current activity rather than a prediction of what may happen, as is the case with genomic studies [16,17,18]. The most commonly used metabolomic techniques are mass spectrometry (MS) and nuclear magnetic resonance (NMR), paired with data processing using advanced computational programmes [18,19]. NMR can measure a broad range of metabolites however it is less sensitive than MS. Gas chromatography (GC) or liquid chromatography (LC) coupled to MS provides excellent separation of molecules within a sample according to their mass-to-charge ratio and analysis at a wide range of concentrations. As MS is more sensitive, it can be utilised to measure more metabolites, but it has a longer analysis time than NMR [19]. Metabolomics has been used to investigate pregnancy related changes in maternal and cord blood, fetal membranes, cervico-vaginal secretions, urine and amniotic fluid; with the potential of identifying metabolic profiles or biomarkers associated with different outcomes, such as PTL [20,21], missed miscarriage [22], diagnosis of exposure to chorioamnionitis in the neonate [23], and the effect of maternal diet on amniotic fluid composition [20,21,24]. However, to our knowledge there are no previous studies focusing on parturition.

Here we present novel MS data from a pilot study designed to investigate the effect of parturition on plasma metabolites in the fetal and maternal circulation at the time of delivery and demonstrate that even with a relatively small sample size significant and robust changes can be identified providing new insights into the metabolic responses of the maternal-feto-placental unit in spontaneous labour.

## 2. Materials and Methods

### 2.1. Participants

Nine women with a vaginal delivery (VD) following active labour of spontaneous onset and ten women who had an elective caesarean section (CS) for reasons not related to maternal or fetal disease gave informed, written consent for inclusion before they participated in the study. The study was conducted in accordance with the Declaration of Helsinki, and ethical approval for the study was gained from the National Research Ethics Service Committee-South West, Bristol (reference number: E5431). The inclusion criteria were: women with uncomplicated singleton term pregnancies; age 18–40 years; no significant past medical history. The exclusion criteria were: multiple pregnancies; age under 18 years or over 41 years of age; taking medications likely to affect metabolomics; diabetes; pre-eclampsia; other metabolic conditions; raised temperature or signs of fetal distress during labour. The following demographic information was collected: ethnicity, age, BMI, smoking status, time last eaten, time of delivery, duration of delivery, drugs given during hospital stay, Apgar scores of the baby, and cord gases if taken.

### 2.2. Collection of Samples

Cord blood and maternal (intervillous) blood samples were obtained within 30 minutes of delivery of the baby and placenta, as described in detail previously [25], following delayed cord clamping which is routinely carried out at St Michael’s Hospital for both VD and elective CS provided there are no contraindications. Samples were collected using a sterile 21 Gauge needle and syringe, transferred into Vacutainer tubes containing EDTA, then centrifuged at 1000× *g* for ten minutes. 200 µL of the clear upper plasma layer was transferred into chilled propylene tubes and stored at −80 °C.

### 2.3. Metabolomic Analysis

The samples were transported on dry ice to Metabolon, Inc. (Morrisville, NC, USA) for ultrahigh performance liquid chromatography-tandem mass spectrometry (UHPLC/MS) [26,27,28]. All samples were analysed together in order to reduce risk of batch bias. This included the measurement of 826 known metabolites for each sample, grouped into 108 sub-pathways and the following eight super-pathways: Lipid, Amino Acid, Peptide, Energy, Nucleotide, Cofactors and Vitamins, and Xenobiotics. Briefly, plasma samples were subjected to methanol extraction then split into aliquots for analysis by UHPLC/MS in the positive (involving two methods, one optimized for hydrophilic, and the other hydrophobic compounds), negative or polar ion mode. Metabolites were identified by automated comparison of ion features to a reference library of chemical standard, followed by visual inspection for quality control [29]. Multiple water blanks were included on each plate of experimental samples to identify any compounds resulting from storage or handling. Compounds which were detected at a level at least triple that found in the water blanks and which were confirmed to be present relative to a chemical reference standard were included in the final analysis. For quality assurance and quality control (QC), pooled QC plasma replicates, as well as several internal standards, were assessed to determine instrument variability, with representative relative standard deviation (RSD) = 3% for internal standards and 7% for endogenous biochemicals.

### 2.4. Statistical Analysis

The two experimental groups were labelled VC/EC and VM/EM. VC/EC represents the difference in the means between VC, the metabolite quantity identified in cord plasma from women who laboured and delivered vaginally, and EC, the metabolite quantity identified in cord plasma from women who did not labour and delivered via elective CS. VM/EM represents the difference in the means between VM, the metabolite quantity identified in intervillous plasma from women who laboured and delivered vaginally, and EM, the metabolite quantity contained in intervillous plasma from women who did not labour and delivered via elective CS. Welch’s two-sample t-test was used to identify metabolites for which the means were significantly different between VC and EC or VM and EM. Analysis was performed on log-transformed data, considered significant if *p* ≤ 0.05. A *q*-value was also calculated to control for false discovery rate and account for the multiple comparisons which occur in metabolomic-based studies, where a *q*-value of <0.10 gives high confidence that a significant difference is not due to chance.

## 3. Results

### 3.1. Demographics

Table 1 presents the characteristics of the labouring and non-labouring groups. All nine women in the vaginal delivery group had confirmed labours of spontaneous onset. All women were of white ethnicity and there were no significant differences in age, parity, employment status, Apgar scores at birth, birthweight or gender of fetus. None of the women had experienced medical complications during the pregnancy, nor had there been concern for any of the fetuses. The mean BMI at booking was 21.6 in the VD group and 25.6 in the elective CS group (*p* = 0.02), due to an outlier in the elective CS group with BMI 38.3. Further analysis was performed and confirmed her inclusion did not affect the spread of results. The median gestation at delivery was 40 weeks in the VD group and 39 weeks and 2 days in the elective CS group (*p* = 0.04), an expected difference as planned CSs are routinely booked between 39 and 40 weeks’ gestation whereas spontaneous deliveries may occur up to three weeks later.

### 3.2. Metabolomics

All the results refer to ten women in the elective CS group and nine women in the VD group. It was not possible to obtain intervillous blood from one placenta, reducing the VM samples to eight. The numbers of metabolites with significant (*p* ≤ 0.05) differences between VM and EM, and VC and EC, in the eight super-pathways are presented in Table 2. Of the 108 metabolite sub-pathways measured, 83 contained metabolites with significant VC/EC and/or VM/EM differences. For the following analysis, xenobiotic metabolites will be considered separately.

### 3.3. Changes in Cord Plasma Metabolome

Appendix A shows the changes in cord plasma (VC/EC) of specific metabolites within the sub-pathways ordered according to the magnitude of fold-change. 120 metabolites significantly increased in the cord plasma of women who laboured, 102 of which also had a significant *q* value (<0.10), indicating a high likelihood that the significant change was not due to chance. These comprised 58 lipid metabolites, 14 amino acid metabolites, nine carbohydrate metabolites, eight nucleotide metabolites, six energy metabolites, three cofactors and vitamins metabolites, two peptide metabolites and two partially characterised molecules. The metabolite with the largest VC/EC fold-change was heme, which had an average 14.30-fold increase in the cord plasma (*p* = 0.024) taken from women who laboured.

Thirty-nine metabolites significantly decreased in the cord plasma of women who laboured (VC/EC), and 27 of these had a significant *q* value: eleven amino acids, six lipids, four peptides, three cofactors and vitamins and three nucleotides.

### 3.4. Changes in Maternal Plasma Metabolome

Appendix A shows the corresponding changes in maternal plasma metabolites (VM/EM). Overall, 165 metabolites significantly increased in labouring women, all of which had a significant *q* value. These included: nine amino acids, seven carbohydrates, three cofactors and vitamins, three energy metabolites, 139 lipids, two nucleotides, and two partially characterised molecules. The metabolite with the greatest significant-fold increase in maternal plasma of women who laboured was the partially characterised molecule glucuronide of C10H18O2 (8), with an average 10.85-fold increase in VM plasma. The dicarboxylate fatty acids maleate and adipate (C6-DC) increased 10.42-fold and 8.06-fold, respectively.

Thirty-six metabolites were significantly decreased in the maternal plasma of women who laboured when compared with women who did not labour (VM/EM), with a corresponding significant *q* value, comprising fifteen amino acids, four cofactors and vitamins, fourteen lipids, two nucleotides and one peptide. Among these were steroid sulphates, including 3-dehydrocholate (*p* = 0.002), 16α-hydroxy DHEA 3-sulfate (*p* = 0.007) and androsteroid monosulfate C_19_H_28_O_4_S (*p* = 0.017), with 0.13, 0.22 and 0.26 fold-changes, respectively.

### 3.5. Xenobiotics

The findings within the xenobiotic metabolites were as expected, reflecting the drugs known to be administered to the women. For example, metronidazole was given to all ten women who were delivered via elective CS but not to any of the women who delivered vaginally, and this was reflected in the results (Appendix A). 

### 3.6. Lipid Super-Pathway

The lipid super-pathway contained the most differences between labouring and non-labouring women (Figure 1g and Table 2). Most of these changes were significant increases in maternal plasma (VM/EM). 405 metabolites were identified in this super-pathway: 34.3% (139/405) of the metabolites significantly increased and 3.5% (14/405) significantly decreased in the maternal plasma of women who laboured when compared to non-labour; correspondingly, 18.0% (73/405) of the metabolites significantly increased in the cord plasma of women who laboured, and 3.0% (12/405) significantly decreased. Of note, 22 sphingolipids (Appendix A) significantly increased in maternal plasma of labouring women with only one corresponding increase found in cord plasma; whereas the sub-pathway with the most significant changes seen in the cord plasma within the lipid super-pathway were fatty acid metabolism (acyl carnitine), long chain fatty acids and polyunsaturated fatty acids (Figure 1g). There were significant increases in the medium and long chain fatty acids in both cord and maternal plasma and there were selective changes in the maternal and fetal circulation for acyl carnitines, acyl cholines, and eicosanoids (12-HETE and 23-HHTrE) (Appendix A). 

Of the steroid metabolites, cholesterol was significantly increased in the maternal plasma of women who laboured (VM/EM) (Appendix A) but did not change in the cord plasma (VC/EC). 17α-hydroxypregnenolone 3-sulfate and 21-hydroxypregnenolone monosulfate were significantly decreased in the maternal plasma of women who laboured (VM/EM), with no corresponding changes seen in the cord plasma (Appendix A). Conversely, progesterone significantly increased in the cord plasma (VC/EC), with no change seen in maternal plasma (VM/EM). There were significant increases in 5α-pregnan-3β-ol,20-one sulfate, 5α-pregnan-3β,20β-diol monosulfate, 5α-pregnan-3β,20α-diol monosulfate, 5α-pregnan-3β,20α-diol monosulfate and pregnanediol-3-glucuronide in maternal plasma, which were not seen in the cord (Appendix A). 

Of the endocannabinoid metabolites, three were significantly increased in the cord plasma of women who laboured when compared with those who did not (VC/EC); and one significantly increased, and one significantly decreased in the maternal plasma of women who laboured (VM/EM) (Appendix A). Twelve of seventeen ceramide metabolites significantly increased in the maternal plasma (VM/EM), however there were no such significant differences seen in the cord plasma samples (VC/EC) (Appendix A). 

Corticosterone significantly increased in maternal plasma (VM/EM) and in cord plasma (VC/EC) (5.41 and 3.09-fold, respectively) of women who laboured, and the corresponding increases for cortisol were 5.02-fold and 3.57-fold. There were also significant increases in cortisone and cortisone 21-sulfate in cord plasma but not in maternal plasma (Appendix A). 

Androstenediol (3β,17β) disulfate and androsteroid monosulfate C_19_H_28_O_4_S significantly decreased in maternal plasma of women who laboured (VM/EM). 16a-hydroxy DHEA 3-sulfate significantly decreased in both the cord and maternal plasma. Androsterone glucuronide, epiandrosterone sulfate, 5α-androstan-3β and 17β-diol disulfate significantly increased in the maternal plasma, and androsterone sulfate and androstenediol (3α, 17α) monosulfate significantly increased in both maternal and cord plasma (Appendix A). 

Of the oestrogenic steroids, estrone 3-sulfate significantly increased in the cord plasma with no change in the maternal plasma, while estriol 3-sulfate significantly decreased in the maternal plasma with no change in the cord (Appendix A).

### 3.7. Nucleotides

The majority of the significant changes seen within the nucleotide super-pathway were within VC/EC (Figure 1d), including significant increases in inosine, hypoxanthine, and xanthine, and a significant decrease in N1-methylinosine. 5-methyluridine (ribothymidine) significantly decreased in both cord (VC/EC) and maternal plasma (VM/EM), while 5,6-dihydrouracil and xanthosine significantly increased (Appendix A). 

### 3.8. Amino Acids

Unlike the other super-pathways, most of the significant changes within the amino-acid super-pathway were decreases in VC/EC and VM/EM (Figure 1a). Serine, histidine, tryptophan, N-acetylkynurenine, picolinate, indole-3-carboxylic acid, citrulline and arginine significantly decreased in both VC/EC and VM/EM. 

Carnosine, N-acetyltyrosine and xanthurenate significantly decreased in VC/EC only, while S-1-pyrroline-5-carboxylate, indolebutyrate, ornithine, homoarginine, threonine, lysine and trans-4-hydroxyproline decreased in VM/EM only. By contrast, N-acetylaspartate increased 7.14-fold in VM/EM only, and cysteine-glutathione disulfide increased 5.21-fold in VC/EC (Appendix A).

### 3.9. Peptides

Twenty metabolites were measured in this super-pathway (Figure 1e and Table 2). Two of the four fibrinogen cleavage peptide metabolites measured were significantly increased in cord plasma of women who laboured compared with non-labour: fibrinopeptide A des-ala increased 12.12-fold and fibrinopeptide A phosphono-ser increased 11.34-fold. There were no corresponding changes in maternal plasma. 

### 3.10. Carbohydrate Metabolism

Twenty nine metabolites were measured in this super-pathway (Figure 1c and Table 2), and of these glucose, maltose, fructose, ribulonate/xylulonate, pyruvate and lactate were significantly increased in both the cord and maternal plasma taken from women who laboured compared with non-labour. 3-phosphoglycerate was significantly increased in cord plasma. 

### 3.11. Energy Super-Pathway

Eleven metabolites were measured in this super-pathway (Table 2). Six metabolites of the TCA cycle were significantly increased in cord plasma of labouring women, but only three increased in maternal plasma (Appendix A). There were no significant changes in oxidative phosphorylation.

### 3.12. Co-Factors and Vitamins

Thirty-one metabolites were measured in this super-pathway (Figure 1b and Table 2). Of these, 1-methylnicotinamide, N1-methyl-2-pyridone-5-carboxamide and N1-methyl-4-pyridone-3-carbox-amide from nicotinate and nicotinamide metabolism were significantly decreased in the cord plasma taken from women who laboured (VC/EC), and adenosine 5’-diphosphoribose (ADP-ribose) significantly increased (Appendix A). There were no such changes in the maternal plasma. 

Heme significantly increased 14.3-fold in cord plasma (VC/EC), with no change seen in the maternal plasma, and two metabolites of vitamin A metabolism and one metabolite of vitamin B6 metabolism were significantly increased with labour in maternal plasma (VM/EM), with no corresponding changes in the cord (Appendix A). 

## 4. Discussion

To our knowledge, this study is unique in its comparison of the metabolomic profile of cord and maternal plasma at the time of delivery between women who have laboured and delivered vaginally and women who have not laboured and delivered via elective caesarean section. The clear-cut differences in metabolite levels between the two groups confirm the strong impact that active labour has on both mother and neonate. A proportion of these changes may be due to the stress of labour itself, but we believe that within these metabolites there are clues as to which pathways trigger spontaneous parturition in women. In addition, the observed changes in xenobiotics such as metronidazole were as expected, supporting the robustness of metabolomic methodology in this setting. While it cannot be ruled out that the intervillous blood samples had some degree of fetal blood contamination, the very distinct metabolic profiles that we obtained in cord and intervillous plasma confirms the fetal and maternal circulations were sampled separately [25], thereby confirming the reliability of this non-invasive method for experimental sampling of maternal blood after delivery.

It could be argued that the stress of labour resembles strenuous exercise, and it would be useful to look to studies which investigate metabolic changes due to exercise to see if they match any of the changes observed here. Medium- and long-chain acetylcarnitines increase up to 9-fold following intense exercise in men [30]. A major source of acetylcarnitine is acetyl Co-A, formed from pyruvate oxidation, and this increase following exercise is likely due to a switch to increased fatty acid oxidation in skeletal muscle instead of glycolysis during exercise [31,32]. The significant increases in acylcarnitine fatty acid metabolism observed in both the cord and maternal plasma of labouring women (VC/EC and VM/EM) (Appendix A) could in part be explained by fatty acid oxidation for energy used by the woman and neonate during labour and delivery. However, while the source of this may be from working skeletal muscle in the labouring mother, this explanation is less likely in the fetus/neonate who is not thought to be using their skeletal muscles during labour and delivery. The increased levels of stress metabolites observed in labour, including cortisol, are likely to be generating fatty acid mobilisation [33]. The increase in sphingomyelins observed in labouring maternal plasma and the increase in acylcarnitine fatty acid metabolism in both the cord and maternal plasma of labouring women could indicate a considerable shift in intermediary metabolism at the time of labour following the relative decrease of these metabolites in pregnancy [34].

Labour and vaginal delivery would be expected to affect metabolites of fetal and maternal adrenal origin, particularly glucocorticoids, as part of the physiological stress response [35,36]. In support of this theory, we found cortisol and its major metabolites cortisone, cortisone 21-sulfate and corticosterone were significantly increased in the cord plasma of women who spontaneously laboured and delivered vaginally (VC) when compared with those who did not labour (EC). Similarly, cortisol and corticosterone were significantly increased in the maternal plasma taken from women who laboured and delivered vaginally (VM) when compared with those who did not labour (EM) (Appendix A). Our data confirm that there is activation of the fetal and maternal adrenal cortex during labour and vaginal delivery, and while this may be part of the mechanism initiating spontaneous labour, further studies are needed to investigate this hypothesis.

Within cord plasma (VC/EC), heme was the metabolite with the greatest fold-change (Appendix A). Heme is a complex of iron with protoporphyrin IX, present in haemoglobin and myoglobin. Its levels in the plasma can increase rapidly following haemolysis or tissue injury [37]. It is possible that the increased heme present in the cord taken from women who laboured is a marker of the increased trauma and tissue damage which a neonate experiences during vaginal delivery when compared with delivery via caesarean section.

### 4.1. Endocannabinoids, Ceramides and Sphingolipids

Endocannabinoids have been investigated as biomarkers for reproductive events, with both predictive and diagnostic uses [38]. Formed from membrane phospholipids, endocannabinoids are described as non-classical neurotransmitters [38]. Of the two characterised cannabinoid receptors, CB1 is present in areas of the brain, peripheral nerve terminals and extraneural sites including the uterus, whereas CB2 is largely limited to cells and organs of the immune system [39]. N-arachidonoylethanolamine (anandamide or AEA) binds to CB1 and CB2 and is broken down by the enzyme Fatty Acid Amide Hydrolase (FAAH) [39]. Cannabis exogenously acts on the endocannabinoid system, and its use has been associated with pregnancy complications such as preterm birth, placental abruption, fetal growth restriction and spontaneous miscarriage [40]. Elevated plasma AEA in early pregnancy (<8 weeks) and lower peripheral lymphocyte expression of FAAH has been associated with miscarriage [40,41], and the CB1-knock out mouse has elevated corticotrophin-releasing hormone (CRH) and spontaneous preterm labour [42]. The uterus can produce its own AEA [39], and myometrial studies show a concentration-dependent relaxation effect of AEA on human myometrial contraction in vitro [43]. Of note, FAAH converts AEA to ethanolamine and releases arachidonic acid for prostaglandin synthesis, providing a potential link between elevated AEA and labour [42]. Ceramide is a ubiquitous sphingolipid second messenger released rapidly in response to apoptosis and stress [44]. CB1 activation induces sphingomyelin hydrolysis and acute production of ceramide, while CB2 activation leads to *de novo* sustained ceramide production [39]. The pro-inflammatory cytokine TNFα is involved in CB1 and CB2 activation and ceramide production [39,44,45], whereby inflammatory events may stimulate the onset of labour.

In this study, AEA, oleoyl ethanolamide (OEA), and palmitoyl ethanolamide (PEA) were significantly increased in the cord plasma taken from the women who laboured (VC/EC) (Appendix A). PEA was significantly increased in the corresponding maternal plasma (VM/EM), while N-oleoylserine significantly decreased in maternal plasma. Interestingly, while in previous studies maternal plasma AEA has been shown to decline throughout pregnancy until a sharp increase at the time of spontaneous or induced labour [40,42], we found no difference in the maternal plasma (VM/EM) immediately following delivery between women who laboured and women who did not; however there was a significant increase in endocannabinoids, including AEA, in the cord plasma of women who laboured (VC/EC). It is possible that any increase in AEA is localised to the placenta during labour, and it has previously been shown that AEA concentrations are significantly higher in the umbilical vein than the umbilical arteries, which suggests either production by or transportation across the placenta [40]. In addition, we found 71% (12/17) of the measured ceramides increased significantly in the maternal plasma (VM/EM) (Appendix A), while none increased in the cord plasma (VC/EC). Correspondingly, 40% (22/55) of measured sphingolipids were increased in the maternal plasma (VM/EM) while only 2% (1/55) significantly changed in the cord plasma (VC/EC). This finding supports previous reports of increased ceramide levels and expression of serine palmitoyl transferase, the rate-limiting enzyme for ceramide synthesis *de novo*, in the placentas of women who laboured compared with those who delivered by elective CS. This group also reported an increase in interleukin-6, suggesting the involvement of ceramide signalling and inflammatory mechanisms in labour [46]. Our data support the view that the endocannabinoid/ceramide/sphingolipid pathway has the potential to stimulate labour [40,42,46]. 

### 4.2. Prostaglandins

Prostaglandins are involved in the early preparation for parturition including cervical ripening (prostaglandin E2) and stimulation of contractility (prostaglandin F2α), as well as rupture of membranes. Increases in prostaglandins, prostacyclin and their metabolites during advancing labour have been demonstrated in maternal plasma, amnion and amniotic fluid [47,48,49]. Prostacyclin and thromboxane metabolites increase in maternal urine in threatened preterm labour [50], and the prostacyclin metabolite 6-ketoprostaglandin F1α increases in maternal urine both during vaginal delivery and elective CS [51]. In our study, the eicosanoid 12-HETE was significantly increased in the maternal plasma of women who laboured and delivered vaginally (VM/EM) (Appendix A), however no prostaglandins were detected. This may be due to the concentrations of prostaglandins in plasma being too low to detect using this metabolomic platform, or because the samples were collected following placental separation with a potentially rapid turnover resulting in these eicosanoid metabolites being excreted into the urine prior to sample collection.

### 4.3. Progesterone and Pregnenolone

Functional progesterone withdrawal has been linked to the onset of labour [42]. Although there were no significant differences in progesterone levels in the cord (VC/EC) or maternal (VM/EM) plasma in our study, there was a significant increase in metabolites of the progestin sub-pathway in the maternal plasma of labouring women (VM/EM). There is a strong positive correlation of 3β- and 16α-hydroxysteroids with gestational age [52]. This reflects maturation of the fetal adrenal zone and potential involvement of these metabolites in the preparation for labour.

Parturition has long been associated with changes in progesterone and pregnenolone metabolism in many species [53,54]. Our findings have uncovered a rich and complex pattern of steroid metabolites in spontaneous labour and vaginal delivery reflected in both maternal and fetal circulations (Appendix A). Of interest is the relative increase in VC/EC of pregnenolone and hydroxypregnenolone sulfates which are mirrored by changes in the opposite direction in the maternal circulation (Appendix A). Moreover, progesterone and pregnandiol metabolites increased in both the maternal and fetal circulations (Appendix A). These changes indicate increased turnover of C21 compounds, which are the first steroids derived from cholesterol through the P450scc and 3β-hydroxysteroid dehydrogenase pathway during labour. We have previously demonstrated changes in cholesterol-transporting lipoproteins in the maternal circulation in women in spontaneous labour [55], confirming the need for high cholesterol turnover during parturition. Our results present, for the first time, a comprehensive picture of C21 metabolites in the maternal and fetal circulations in relation to labour and vaginal delivery that can be used to guide future studies investigating steroid changes at parturition.

### 4.4. Estrogen

Estrogen synthesis in pregnancy results from the integration of maternal, placental and fetal tissues [56]. Placental Corticotrophin Releasing Hormone (CRH) stimulates the fetal adrenals to produce DHEA-sulfate (DHEAS), which is 16-hydroxylated in the fetal liver and converted to estriol and estrone in the placenta [57,58]. In our study, 16a-hydroxy DHEA 3-sulfate significantly decreased in both the cord (VC/EC) and maternal plasma (VM/EM), while estriol 3-sulfate significantly decreased in the maternal plasma (VM/EM) and estrone 3-sulfate significantly increased in the cord plasma (VC/EC) of women who laboured (Appendix A). This provides an interesting insight into the effect of labour on the metabolism of estrogens. It is likely that these changes are related to substrate availability, including DHEA and its sulfated derivatives, as well as local changes in enzyme activity involving sulfatases, sulfotransferases, hydroxylases and aromatases, and their redox cofactors including NADH and NADPH in the uterus, placenta, adrenals, liver and other organs. The regulatory effect of hormones and growth factors such as CRH [58,59] and epidermal growth factor [60] on each pathway will be modulated by acute changes in cofactor availability during the process of labour which places strong energy and redox demands on the uterus. Previously reported increases in the estriol/estradiol ratio in maternal plasma occur gradually over several weeks before labour onset [59]. Our findings of a decrease in 16a-hydroxy DHEA 3-sulfate in the cord and maternal plasma and a decrease in estriol 3-sulfate in the maternal plasma of women who laboured may indicate a sudden decrease in production of DHEAS by the fetal adrenals at the time of or during spontaneous labour. This adds further depth to our knowledge of the complex steroid pathways involved in the spontaneous onset of human labour. In addition, changes in estrogenic metabolites in the maternal and fetal circulations may not necessarily reflect local tissue changes, such as within the myometrium, which may also have a functional effect on uterine contractility [61]. While there are many practical advantages in sampling the peripheral circulation, and our study provides clear cut overall differences in steroid metabolites between labouring and non-labouring women, the limitation of metabolomic studies using peripheral blood is that the dilution effect and the contribution of many organs to steroid metabolism may obscure local changes in individual tissues. Further work is necessary to understand these mechanisms, combining peripheral plasma metabolomics with measurements of enzyme activity in specific tissues.

A potential limitation to this study is the relatively low number of participants in each group. However, clear and significant changes in metabolite levels were identified within this pilot group with relatively little noise, demonstrating the reliability of this approach. The platform used is very robust with 26.9% (222/826) of metabolites significantly changing in maternal plasma and 21.1% (174/826) in cord plasma with labour. The method is unlikely to show false positive results: for example metronidazole was given at the time of knife-to-skin for all women in the elective CS group, but none of the women in the VD group received it; reassuringly, all the maternal and cord plasma of the non-labour group contained metronidazole and none of the labouring group contained it (Appendix A). Another limitation is the fact that the study is cross sectional, rather than serial, so it is difficult to know whether the changes observed with labour are part of the mechanism of the initiation of labour or a consequence of the stress and metabolic demands of several hours of active labour. Nevertheless, our data show that metabolomic techniques are a powerful approach to identify biochemical changes associated with labour.

## 5. Conclusions

In this pilot study we have demonstrated a wide range of metabolic changes at the time of delivery between women who laboured spontaneously and women who did not labour. The data will be useful to investigate the biochemical pathways involved in the physiological mechanism of labour in normal term pregnancy and provide a pattern against which pathological changes in preterm labour can be compared. Our results demonstrate an interplay between metabolites in the maternal and fetal circulations which point to the involvement of the endocannabinoid, sphingolipid, ceramide and steroid systems in the mechanism of active labour. Further studies are now warranted, where serial maternal samples are taken throughout pregnancy to determine when these changes take place, especially focusing on the weeks and days immediately prior to spontaneous labour.

## Figures and Tables

**Figure 1 ijerph-16-01527-f001:**
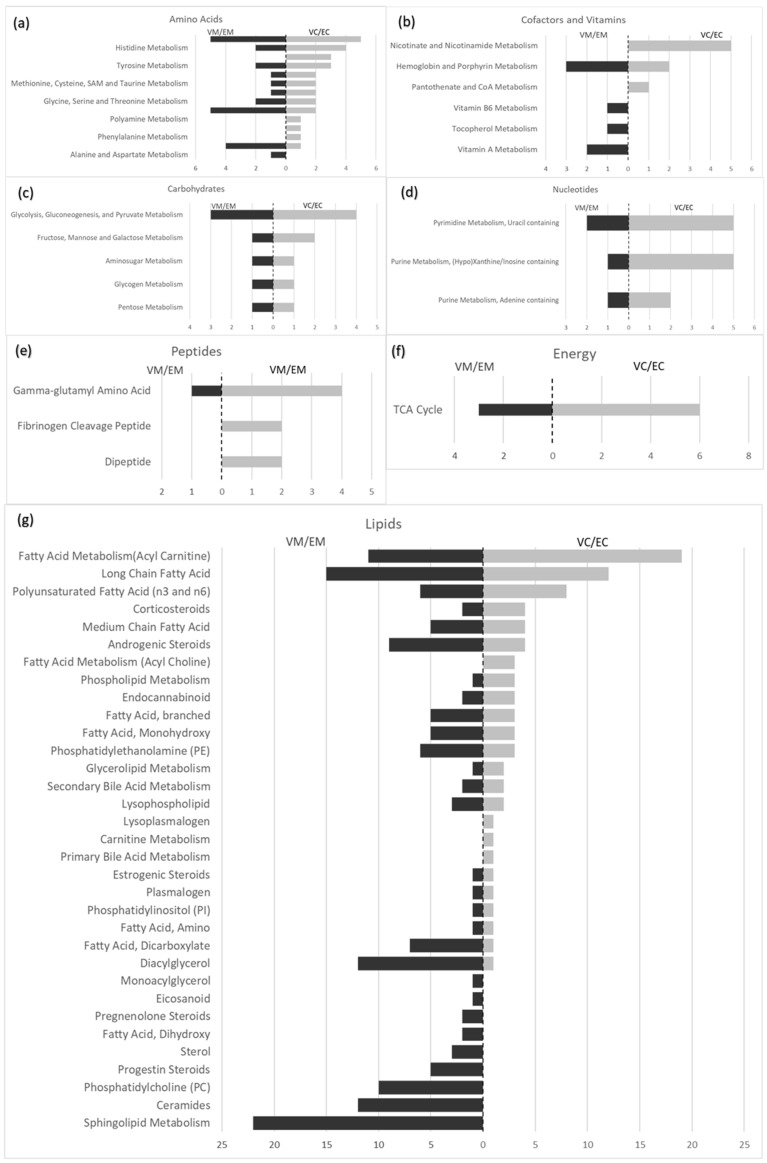
Comparative histograms showing the number of metabolites with significant changes between maternal (intervillous) plasma taken from women who laboured and maternal plasma (intervillous) taken from women who did not labour (VM/EM) compared with cord plasma taken from women who laboured and cord plasma taken from women who did not labour (VC/EC) for each if the sub-pathways within the following super-pathways: (**a**) Amino Acids; (**b**) Cofactors and Vitamins; (**c**) Carbohydrates; (**d**) Nucleotides; (**e**) Peptides; (**f**) Energy; (**g**) Lipids.

**Table 1 ijerph-16-01527-t001:** Characteristics of included human participants.

Demographic	VD n = 9	Elective CS n = 10	*p* Value (at ≤ 0.05)
Age (years)	32.4 (24–39) ^a^	34.6 (30–40) ^a^	NS
BMI at pregnancy booking	21.6 (19.9–23.1) ^a^	25.56 (21.7–38.3) ^a^	0.02
Ethnicity	White British: 9	White British: 9 White Romanian: 1	NS
Smoking history:			NS
Never	8	4	
Ex	0	5	
Current	1 (4/day)	1 (10/day)	
Maternal past medical history	Eczema: 1 ICSI pregnancy (own sperm/egg): 1	Previous postnatal depression:1 Well controlled asthma: 1 (MTHFR gene 1): 1	NA
Maternal employment status at pregnancy booking:			NS
Employed	8	8	
Home maker	1	2	
Gravida	1.2 (0–2) ^a^	2.4 (0–5) ^a^	NS
Parity	0.6 (0–2) ^a^	1.7 (0–7) ^a^	NS
Significant pregnancy complications	Nil	Nil	NA
Indication for elective CS	NA	Breech: 3 Previous CS: 3 Maternal request: 1 Previous 3rd degree vaginal tear: 1 Previous traumatic delivery: 1 Tocophobia: 1	NA
Duration of labour in minutes	312 min (70–650) ^a^	NA	NA
Gestation at delivery (weeks+days)	40 (38+1–41+6) ^a^	39+2 (38+4–40+3) ^a^	0.04
Apgar scores at 1, 5 and 10 min:			NS
8,9,10	1	0	
9,10,10	8	10	
Management of third stage:			NS
Syntometrine or Carbetocin	8	10	
Physiological	1	0	
Interval between delivery and freezing of sample (minutes)	38 (36–60) ^b^	28.5 (24–45) ^b^	0.02
Gender of fetus:			NS
Female	5	4	
Male	4	6	
Apgar scores at 1, 5 and 10 min:			NS
8,9,10	1	0	
9,10,10	8	10	
Birth weight (Kilograms):	3.4 (3.2–3.9) ^a^	3.6 (3.1–4.1) ^a^	NS

^a^ Mean (range); ^b^ Median (range); VD = Vaginal Delivery; CS = Caesarean Section; BMI = Body Mass Index; NS = not significant; NA = non-applicable; ICSI = intracytoplasmic sperm injection.

**Table 2 ijerph-16-01527-t002:** Number of metabolites which significantly increased or decreased at the time of delivery between women who spontaneously laboured and delivered vaginally and women who did not labour and delivered via elective caesarean section in the maternal (intervillous) plasma (VM/EM) and cord plasma (VC/EC) for each of the super pathways.

Super Pathway	Number Metabolites Measured	Number Significant Changes VM/EM			Number Significant Changes VC/EC		
		↑	↓	Total (%)	↑	↓	Total (%)
**Lipid**	405	139	14	153 (37.8)	73	12	85 (21.0)
**Amino Acid**	177	9	15	24 (13.6)	15	14	29 (16.4)
**Xenobiotics**	108	18	3	21 (19.4)	16	5	21 (19.4)
**Carbohydrate**	29	7	0	7 (24.1)	9	0	9 (31.0)
**Cofactors and Vitamins**	31	3	4	7 (22.6)	3	5	8 (25.8)
**Energy**	11	3	0	3 (27.3)	6	0	6 (54.6)
**Nucleotide**	38	2	2	4 (10.5)	9	3	12 (31.6)
**Peptide**	24	0	1	1 (4.2)	3	5	8 (33.3)
**Partially characterised**	3	2	0		2	0	

VM = Maternal (intervillous) plasma from women who laboured and delivered vaginally (n = 8); EM = maternal plasma from women who did not labour and delivered via caesarean section (n = 10); VC = cord plasma from women who laboured and delivered vaginally (n = 9); EC= cord plasma from women who did not labour and delivered via elective caesarean section (n = 10); ↑ = Number of metabolites significantly elevated; ↓ = Number of metabolites significantly decreased.

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
