# Peer review of "Metabolite Changes in Maternal and Fetal Plasma Following Spontaneous Labour at Term in Humans Using Untargeted Metabolomics Analysis: A Pilot Study"

_ijerph, 2019, doi:10.3390/ijerph16091527_

Round 1
Reviewer 1 Report
The authors should be congratulated for an excellent study in an area that innovation is needed to understand the complex biologic processes that lead to labour.
One specific question is whether the metabolomics testing included prostaglandin or its metabolites.
Specific comments
Line 330, PEA is listed, but this reviewer could not find what it means.
betahrydoxysteroid —> betahydroxysteroid ?
The supplemental tables provide an impressive list of compounds that were detected, but there is no mention of the detection level for each compound and how below the level of detection was handled.
How was the statistical analysis using t-test done when one group was zero or below the level of detection and the other sample had measurable levels?
On table 1, the authors use mean for Interval between delivery and freezing of sample (minutes). Seems that given the small sample size, median may be more appropriate.
Author Response
Response to Reviewer 1 Comments
Thank you very much for your comments and suggestions. Please find our responses below.
Point 1: One specific question is whether the metabolomics testing included prostaglandin or its metabolites.
Response 1: Thank you. We have now added the below paragraph regarding prostaglandin in the discussion section (355-366):
4.2. Prostaglandins
Prostaglandins are involved in the early preparation for parturition including cervical ripening (prostaglandin E2) and stimulation of contractility (prostaglandin F2alpha), as well as rupture of membranes. Increases in prostaglandins, prostacyclin and their metabolites during advancing labour have been demonstrated in maternal plasma, amnion and amniotic fluid [47-49]. Prostacyclin and thromboxane metabolites increase in maternal urine in threatened preterm labour [50], and the prostacyclin metabolite 6-keto-prostaglandin F1 alpha increases in maternal urine both during vaginal delivery and elective CS [51]. In our study, the eicosanoid 12-HETE was significantly increased in the maternal plasma of women who laboured and delivered vaginally (VM/EM) (Table S2), however no prostaglandins were detected. This may be due to the samples being collected following placental separation with a potentially rapid turnover resulting in these eicosanoid metabolites being excreted into the urine prior to sample collection.
Point 2: Line 330, PEA is listed, but this reviewer could not find what it means
Response 2: Thank you. PEA is Palmitoyl Ethanolamide (Table S8). We have now clarified this within the manuscript (line 334).
Point 3: The supplemental tables provide an impressive list of compounds that were detected, but there is no mention of the detection level for each compound and how below the level of detection was handled.
Response 3: Thank you. We have now addressed this within the main manuscript (lines 97-101). In further detail, given the wide range of ionisation efficiencies of molecules across all biochemical classes, this precludes assessment of a limit of detection provided in a targeted panel with multi-point calibration. For the untargeted, global platform, Metabolon, Inc. analyses multiple water blanks on each plate of experimental samples to identify those compounds present in the sample resulting from storage, handling and analysis. Compounds that are present at greater than triple the water blank levels and confirmed to be present relative to a chemical reference standard are included in the final data for analysis.
Point 4: How was the statistical analysis using t-test done when one group was zero or below the level of detection and the other sample had measurable levels?
Response 4: Data sets were imputed with the minimum value across all samples for each metabolite, providing a statistical floor for pairwise comparisons.
Point 5: On table 1, the authors use mean for Interval between delivery and freezing of sample (minutes). Seems that given the small sample size, median may be more appropriate.
Response 5: Thank you. We have now amended this accordingly in the manuscript (Table 1).
Reviewer 2 Report
The study by Birchenall et al provides a comprehensive biochemical picture of metabolites in maternal and fetal circulation at the time of delivery (spontaneous term labor and elective C-section) which serves as a standard against which pathological changes in pregnancy can be compared.
It is expected from the authors to discuss fetal gender had an impact on the metabolites and analysis.
Author Response
Response to Reviewer 2 Comments
Thank you very much for your comments and suggestion. Please find our response below.
Point 1: It is expected from the authors to discuss fetal gender had an impact on the metabolites and analysis.
Response 1: Thank you. We have now added fetal gender to Table 1 within the manuscript. There were 5 females and 4 males in the vaginal delivery group and 4 females and 6 males within the elective caesarean section group. This was non-significant. We have added this finding to the text of the results section (line 123).
Reviewer 3 Report
In the current study, Birchenall et al. have investigated the effect of parturition on plasma metabolites in the fetal and maternal circulation at the time of delivery and demonstrate the involvement of endocannabinoid, sphingolipid, ceramide and steroid systems in the mechanism of active labor. Overall, this is a well-designed study with appropriate statistical analyses. However, there are a few questions that the authors need to address and the manuscript can be considered for publication.
Comments for the authors:
1. The authors need to include fetal characteristics and outcome in the Table 1. Also, if there are any disparities in the socioeconomic status of the subjects enrolled, it should be included in the current study.
2. In the current study, AEA, OEA, and PEA were significantly increased in the cord plasma taken from the women who labored (VC/EC) (Table S8). The authors need to explain as to why the AEA levels did not increase in VM/EM. In the current study did the authors find any changes in the arachidonic acid and prostaglandins, as prostaglandins play a critical role in the initiation of human labor (Lee DC et al., J Clin Endocrinol Metab. 2010).
3. The authors have discussed adequately about progesterone and its metabolites in the manuscript. However, they need to discuss in detail about the estrogenic metabolites and their role in labor. In humans, during parturition there appears to be a change in the ratio of the E2 and E3 as labor approaches, leading to 10 fold excess of E3 (Smith R et al., J Clin Endocrinol Metab. 2009). Recently, it has been shown that a change in estrogen receptors at term has been linked to the onset of labor (Anamthathmakula P et al., EBioMedicine 2019; Smith R et al., EBioMedicne 2019) similar to the functional progesterone withdrawal due to the differential expression of the progesterone receptor isoforms (Merlino AA et al., J Clin Endocrinol Metab. 2007). The authors need to discuss in detail why estriol 3-sulfate was low in VM/EM while estrone 3-sulfate is high in VC/EC? The authors report that 16a-hydroxy DHEA 3-sulfate significantly decreased in both the cord and maternal plasma. During pregnancy, estriol is derived from Dehydroepiandrosterone sulphate (DHEAS), which is produced from the fetal adrenal under the stimulation of placental Corticotrophin Releasing Hormone. Does the decrease in DHEAS be responsible for the decrease in estriol levels in VM/EM?
Author Response
Response to Reviewer 3 Comments
Thank you very much for your comments and suggestions. Please find our responses below.
Point 1: The authors need to include fetal characteristics and outcome in the Table 1. Also, if there are any disparities in the socioeconomic status of the subjects enrolled, it should be included in the current study.
Response 1: Thank you. We have now added fetal gender, birthweight and maternal employment status at booking to Table 1 within the manuscript. All were non-significant between the two experimental groups and this finding has been added to the text of the main manuscript within the results section (lines 122-123).
Point 2: In the current study, AEA, OEA, and PEA were significantly increased in the cord plasma taken from the women who labored (VC/EC) (Table S8). The authors need to explain as to why the AEA levels did not increase in VM/EM. In the current study did the authors find any changes in the arachidonic acid and prostaglandins, as prostaglandins play a critical role in the initiation of human labor (Lee DC et al., J Clin Endocrinol Metab. 2010).
Response 2: Thank you. We have now extended the discussion regarding AEA (lines 337-353); and have added a paragraph regarding prostaglandins to the discussion section (lines 355-366). The added text is as follows:
Lines 337-353:
Interestingly, while in previous studies maternal plasma AEA has been shown to decline throughout pregnancy until a sharp increase at the time of spontaneous or induced labour [40,42], we found no difference in the maternal plasma (VM/EM) immediately following delivery between women who laboured and women who did not; however there was a significant increase in endocannabinoids, including AEA, in the cord plasma of women who laboured (VC/EC). It is possible that any increase in AEA is localised to the placenta during labour, and it has previously been shown that AEA concentrations are significantly higher in the umbilical vein than the umbilical arteries, which suggests either production by or transportation across the placenta [40]. In addition, we found 71% (12/17) of the measured ceramides increased significantly in the maternal plasma (VM/EM) (Table S9), while none increased in the cord plasma (VC/EC). Correspondingly, 40% (22/55) of measured sphingolipids were increased in the maternal plasma (VM/EM) while only 2% (1/55) significantly changed in the cord plasma (VC/EC). This finding supports previous reports of increased ceramide levels and expression of serine palmitoyl transferase, the rate-limiting enzyme for ceramide synthesis de novo, in the placentas of women who laboured compared with those who delivered by elective CS. This group also reported an increase in interleukin-6, suggesting the involvement of ceramide signalling and inflammatory mechanisms in labour [46]. Our data support the view that the endocannabinoid/ceramide/sphingolipid pathway has the potential to stimulate labour [40,42,46].
Lines 355-366:
4.2. Prostaglandins
Prostaglandins are involved in the early preparation for parturition including cervical ripening (prostaglandin E2) and stimulation of contractility (prostaglandin F2alpha), as well as rupture of membranes. Increases in prostaglandins, prostacyclin and their metabolites during advancing labour have been demonstrated in maternal plasma, amnion and amniotic fluid [47-49]. Prostacyclin and thromboxane metabolites increase in maternal urine in threatened preterm labour [50], and the prostacyclin metabolite 6-keto-prostaglandin F1 alpha increases in maternal urine both during vaginal delivery and elective CS [51]. In our study, the eicosanoid 12-HETE was significantly increased in the maternal plasma of women who laboured and delivered vaginally (VM/EM) (Table S2), however no prostaglandins were detected. This may be due to the samples being collected following placental separation with a potentially rapid turnover resulting in these eicosanoid metabolites being excreted into the urine prior to sample collection.
Point 3: The authors have discussed adequately about progesterone and its metabolites in the manuscript. However, they need to discuss in detail about the estrogenic metabolites and their role in labor. In humans, during parturition there appears to be a change in the ratio of the E2 and E3 as labor approaches, leading to 10 fold excess of E3 (Smith R et al., J Clin Endocrinol Metab. 2009). Recently, it has been shown that a change in estrogen receptors at term has been linked to the onset of labor (Anamthathmakula P et al., EBioMedicine 2019; Smith R et al., EBioMedicne 2019) similar to the functional progesterone withdrawal due to the differential expression of the progesterone receptor isoforms (Merlino AA et al., J Clin Endocrinol Metab. 2007). The authors need to discuss in detail why estriol 3-sulfate was low in VM/EM while estrone 3-sulfate is high in VC/EC? The authors report that 16a-hydroxy DHEA 3-sulfate significantly decreased in both the cord and maternal plasma. During pregnancy, estriol is derived from Dehydroepiandrosterone sulphate (DHEAS), which is produced from the fetal adrenal under the stimulation of placental Corticotrophin Releasing Hormone. Does the decrease in DHEAS be responsible for the decrease in estriol levels in VM/EM?
Response 3: Thank you. We have added a section regarding estrogenic metabolites to the discussion section (lines 388-417). The added text is as follows:
4.4. Estrogen
Estrogen synthesis in pregnancy results from the integration of maternal, placental and fetal tissues [56]. Placental Corticotrophin Releasing Hormone (CRH) stimulates the fetal adrenals to produce DHEA-sulfate (DHEAS), which is 16 hydroxylated in the fetal liver and converted to estriol and estrone in the placenta [57,58]. In our study, 16a-hydroxy DHEA 3-sulfate significantly decreased in both the cord (VC/EC) and maternal plasma (VM/EM), while estriol 3-sulfate significantly decreased in the maternal plasma (VM/EM) and estrone 3-sulfate significantly increased in the cord plasma (VC/EC) of women who laboured (Tables S1 and S2). This provides an interesting insight into the effect of labour on the metabolism of estrogens. It is likely that these changes are related to substrate availability, including DHEA and its sulfated derivatives, as well as local changes in enzyme activity involving sulfatases, sulfotransferases, hydroxylases and aromatases, and their redox cofactors including NADH and NADPH in the uterus, placenta, adrenals, liver and other organs. The regulatory effect of hormones and growth factors such as CRH [58,59] and epidermal growth factor [60] on each pathway will be modulated by acute changes in cofactor availability during the process of labour which places strong energy and redox demands on the uterus. Previously reported increases in the estriol/estradiol ratio in maternal plasma occur gradually over several weeks before labour onset [59]. Our findings of a decrease in 16a-hydroxy DHEA 3-sulfate in the cord and maternal plasma and a decrease in estriol 3-sulfate in the maternal plasma of women who laboured may indicate a sudden decrease in production of DHEAS by the fetal adrenals at the time of or during spontaneous labour, resulting in the observed decrease in estriol 3-sulfate. This adds further depth to our knowledge of the complex steroid pathways involved in the spontaneous onset of human labour. In addition, changes in estrogenic metabolites in the maternal and fetal circulations may not necessarily reflect local tissue changes, such as within the myometrium, which may also have a functional effect on uterine contractility [61]. While there are many practical advantages in sampling the peripheral circulation, and our study provides clear cut overall differences in steroid metabolites between labouring and non-labouring women, the limitation of metabolomic studies using peripheral blood is that the dilution effect and the contribution of many organs to steroid metabolism may obscure local changes in individual tissues. Further work is necessary to understand these mechanisms, combining peripheral plasma metabolomics with measurements of enzyme activity in specific tissues.